# Development of a Real-Time Knee Extension Monitoring and Rehabilitation System: Range of Motion and Surface EMG Measurement and Evaluation

**DOI:** 10.3390/healthcare10122544

**Published:** 2022-12-15

**Authors:** Kiattisak Sengchuai, Chinnakrit Kanjanaroat, Jermphiphut Jaruenpunyasak, Chonnanid Limsakul, Watcharin Tayati, Apidet Booranawong, Nattha Jindapetch

**Affiliations:** 1Department of Electrical Engineering, Faculty of Engineering, Prince of Songkla University, Songkhla 90110, Thailand; 2Department of Biomedical Sciences and Biomedical Engineering, Faculty of Medicine, Prince of Songkla University, Songkhla 90110, Thailand; 3Department of Rehabilitation Medicine, Faculty of Medicine, Prince of Songkla University, Songkhla 90110, Thailand; 4Physical Therapy Unit, Department of Rehabilitation, Trang Hospital, Trang 92000, Thailand

**Keywords:** knee extension, rehabilitation, real-time monitoring, surface EMG, ROM

## Abstract

In this paper, a real-time knee extension monitoring and rehabilitation system for people, such as patients, the elderly, athletes, etc., is developed and tested. The proposed system has three major functions. The first function is two-channel surface electromyography (EMG) signal measurement and processing for the vastus lateralis (VL) and vastus medialis (VM) muscles using a developed EMG device set. The second function is the knee extension range of motion (ROM) measurement using an angle sensor device set (i.e., accelerometer sensor). Both functions are connected and parallelly processed by the NI-myRIO embedded device. Finally, the third function is the graphical user interface (GUI) using LabVIEW, where the knee rehabilitation program can be defined and flexibly set, as recommended by physical therapists and physicians. Experimental results obtained from six healthy subjects demonstrated that the proposed system can efficiently work with real-time response. It can support multiple rehabilitation users with data collection, where EMG signals with mean absolute value (MAV) and root mean square value (RMS) results and knee extension ROM data can be automatically measured and recorded based on the defined rehabilitation program. Furthermore, the proposed system is also employed in the hospital for validation and evaluation, where bio-feedback EMG and ROM data from six patients, including (a) knee osteoarthritis, (b) herniated disc, (c) knee ligament injury, (d) ischemic stroke, (e) hemorrhagic stroke, and (f) Parkinson are obtained. Such data are also collected for one month for tracking, evaluation, and treatment. With our proposed system, results indicate that the rehabilitation people can practice themselves and know their rehabilitation progress during the time of testing. The system can also evaluate (as a primary treatment) whether the therapy training is successful or not, while experts can simultaneously review the progress and set the optimal treatment program in response to the rehabilitation users. This technology can also be integrated as a part of the Internet of Things (IoT) and smart healthcare systems.

## 1. Introduction

Rehabilitation is the care provided to people who have lost their ability to perform daily life activities. Some of the most common causes are injuries and trauma, strokes, major surgery, severe infections, medical treatment side effects, certain birth defects and genetic disorders, disabilities, chronic pain, etc., [1]. Thus, monitoring and rehabilitation via physical therapy are necessary for enhancing rehabilitation people’s quality of life. As the number of people who need rehabilitation increases, the demand for therapy also increases [2].

The main obstacles to effective physical therapy are: lack of consistent participation due to insufficient training by experts (i.e., physicians and physical therapists); lack of motivation to practice; being unable to monitor, track, and control the rehabilitation program when patients leave the hospital; and inconveniences such as travel. Additionally, the number of efficient tools, as well as physical therapists and physicians, is not enough. Exercise for physical rehabilitation often has to be done where patients can be monitored and under continual professional supervision. However, the rehabilitation activities may be done outside of the hospital if the right monitoring is provided. That could decrease the length of rehabilitation time and the frequency of injuries during off-hospital rehabilitation. Another benefit is the ability to quantify and evaluate the effectiveness of each patient’s rehabilitation process [3]. In particular, in a COVID pandemic situation at present, the amount of hospital-based therapeutic exercise decreases significantly due to social distancing and hospital safety policies [4]. As a result, to meet this requirement, some of them would rather do the exercise through a telemedicine system [5].

In Thailand, according to the information of the Physical Therapy Council [6,7], there are 12,000 physical therapists registered for physical therapy. However, approximately 6000 physical therapists still work in both public and private health systems. In community hospitals (735 hospitals in Thailand), there are only 300 community hospitals that have physical therapists, and, on average, only 1.5 physical therapists have worked at each community hospital. It is not enough to service the rehabilitation needs of people both in hospitals and at their homes. In fact, 2 to 4 physical therapists are required for each hospital in practice. In addition, the report also shows that there are many people suffering from osteoarthritis and that the trend is to increase every year. Here, knee osteoarthritis patients have pain in their knee joints during knee movement. To prevent and reduce pain in knee joints, osteoarthritis doctors and physical therapists are required to handle and treat patients also. According to the data from Songklanagarind Hospital, it was found that more than two thousand patients with osteoarthritis were served at the orthopedic clinic. Knee osteoarthritis patients find it difficult to walk and may become unable to help themselves. Consequently, osteoarthritis is one of the most important diseases to be properly treated. Treatment of osteoarthritis is multidisciplinary and depends on the stage of degenerative changes. One of the important treatments for osteoarthritis is rehabilitation and exercise to strengthen the knee muscles. We note that knee osteoarthritis is one of the important knee joint movement problems. There are still other diseases and symptoms related to knee joint movement problems, and related patients need rehabilitation treatments.

According to the requirements and significance of the research problems mentioned above, appropriate monitoring and rehabilitation systems to assist knee joint movement patients both in hospitals and at home are required to be deployed. Moreover, as mentioned above, due to social distancing and hospital safety policies, the amount of hospital-based exercise rehabilitation decreases significantly during the COVID pandemic situation. Thus, some patients alternatively prefer to perform the exercise in their home. Because of advancements in sensing techniques, lower-cost integrated circuits, and the advancement of networking technologies, an autonomous and efficient monitoring and rehabilitation system to assist patients can be established [8]. The review of the related works related to knee rehabilitation systems presented in the literature is provided below. Here, video camera-based technology, wearable sensors, ROM, and EMG measurement related to knee rehabilitation systems are introduced. A summary comparison between related works and our work is also given in Table 1.

**Video camera based**: The authors in [9] verified the usage of the Microsoft Kinect for human motion analysis systems. The dynamic posture control tests were performed by ten healthy volunteers, while their lower limb kinematics were assessed using a standard motion capture system and a single Kinect sensor. The authors claimed that this simple-to-implement system might provide clinicians with a simple tool for analyzing reach distances while developing an understanding of lower movement patterns in healthy and injured populations. Stroke rehabilitation based on an intelligence interaction system was presented by [10]. The authors emphasized that the purpose of rehabilitation training for stroke patients is to minimize physical function degradation and prevent associated problems. Furthermore, traditional rehabilitation equipment is quite costly. Thus, the authors developed a rehabilitation system based on the use of a 3D camera from Intel RealSense D415 to identify the patient’s gesture; a technique for detecting patients’ motions while lying in bed was developed. The findings demonstrated that the system had good monitoring of stroke rehabilitation, because the software could precisely count arm flexion gesture treatment. [11] observed patient movement (i.e., with the Timed-Up-and-Go test: TUG test) using computer vision with a standard RGB camera. Thirty healthy subjects were examined utilizing a Kinect and a regular video feed while doing repetitions of the mobility test at 3 and 1.5 m. The participants’ global 3D postures were then extracted using a mask regional convolutional neural network technique and a deep multitask human sensing architecture. The authors claimed that their results might facilitate the investigation of automating other clinical procedures that were unachievable because of technology restrictions.

**Wearable Sensors**: The clinical usefulness of measuring motor function in post-acute stroke patients was described in [12]. The authors stated that information on everyday living performance is typically unavailable, but it may increase therapeutic personalization. Therefore, sensor-derived information on daily living performance is useful for counseling and program design for stroke patients who live at home. In [12], fifteen stroke patients were examined, and their daily motor performance was evaluated using body-worn inertial sensors connected to the trunk, shanks, and wrists, which calculated physical activity as well as other measures of walking and arm activity. Here, a 3-axis gyroscope, a 3-axis accelerometer, a 3-axis magnetometer, and a barometer measure activities such as laying, sitting, standing, and walking. The authors stated that data on daily motor function can be useful for adapting rehabilitation programs and addressing to the requirements of stroke patients.

**ROM Measurement**: The authors of [13] developed and verified a sensor fusion technique to predict lower-limb joint angles. Two inertial measurement units (or IMUs) were connected below and above the knee joint, and the gyroscope and accelerometer information were gathered. Both the IMU technology and their algorithm could accurately estimate the knee joint angle. A sensory motion system was created by [14] to identify gait abnormalities by estimating angular deviations in the knee and ankle joints. An accelerometer, gyroscope, and magnetometer were utilized to measure the angular motions. The authors concluded that their technology can be used by those who have gait issues to assist them improve their motions.

Ref. [15] reported ROM assessment for robot-assisted ankle-foot rehabilitation. The authors proposed a method to increase the ROM precision of a robotic ankle-foot rehabilitation system by integrating proxy-based sliding mode control and therapist-joined zero-torque control. Ten volunteers, eight males and two females, were recruited for this investigation to examine the method’s feasibility and effect on ankle stretching. The findings showed that the method was capable of improving the measured ROM accuracy. The authors of [16] investigated how sensor location affects the accuracy of knee ROM in a rehabilitation system. An investigation was carried out in a laboratory with healthy persons in a sitting position. Data on knee ROM were collected using the BPMpathway system and a BIOPAC twin-axis digital goniometer at the same time. The authors concluded that the manufacturer should reevaluate the suggested sensor position for knee exercises. When the sensor was positioned below the knee, the error between the BPMpathway system and the contemporaneous system employed (BIOPAC) was the lowest.

**EMG and EEG Measurement**: Since high-heeled shoes may contribute to the development of knee pain, the influence of shoe heel height on EMG activity of VL and VM muscles during sit to stand was examined in [17]. Twenty-five healthy females were evaluated with heel wedges of 1 to 5 cm in height, and EMG activities were recorded during the exercise. The experiments showed that as heel height increased, VL and VM EMGs increased, but there was no difference in the relative EMG intensity of VL and VM as evaluated by the VL to VM ratio.

The authors of [18] developed an EMG-controlled knee exoskeleton to support stroke patients in their rehabilitation. An EMG sensor using Myo with eight active channels (i.e., Ch1 to Ch8) collected the subject’s EMG signals, which were processed by a Kalman filter to drive the exoskeleton automatically. The experiment conducted on six healthy subjects demonstrated that the subjects could utilize the EMG for exoskeleton control to assist them in playing the game. It has the potential to boost the intensity of therapy as well as the outcomes for stroke patients in the early phases of recovery. [19] presented a low-cost EMG system based on the Arduino Mega board and an EMG muscle sensor. The evaluation was carried out through a series of experiments in which subjects performed isometric and dynamic exercises while EMGs of the rectus femoris muscle were recorded using both the suggested system and a commercial system. A rehabilitation strategy based on a lower-limb exoskeleton combined with a human-machine interface (HMI) was reported in [20]. HMI was utilized to capture multimodal signals obtained from a foot motor imagery-based brain–machine interface based on electroencephalographic (EEG) and multichannel EMG signals obtained from leg muscles. The method was examined in healthy participants, and the findings shown that the system could assess multiple signals in real-time. The authors stated that, depending on individual demand, neuromuscular disease, and level of rehabilitation therapy, an integration of EEG and EMG data can be utilized in clinical rehabilitation protocols for robotic exoskeleton usage.

**ROM and EMG Measurement**: A technique for measuring knee joint resistance torque during passive motion utilizing an isokinetic machine (i.e., the Biodex) was introduced in [21]. Based on the angle, angular velocity, and torque output from the Biodex, the authors suggested a solution to adjust for torque due to gravity and inertia. To determine the active torque due to muscle contraction, the authors also measured the EMG and installed a potentiometer on the driving arm of the Biodex to synchronize the EMG signals with the Biodex output. Hence, both ROM and EMG could be monitored simultaneously.

In [22], the authors investigated the association between isometric force and surface EMG of quadriceps femoris muscles in single-joint knee extension (KE) and multi-joint leg push (LP) exercises. Nine healthy males performed actions at a knee angle of 90° at 20, 40, 60, 80, and 100% of maximum contraction. The EMGs of the VL, VM, biceps femoris (BF), and rectus femoris (RF) muscles were collected. Results indicated that all the muscles under investigation appeared to have a similar EMG/force relationship during isometric single-joint KE and multi-joint LP at a 90° angle to the knee. In addition, whereas VM and RF showed linearity deviations in both exercises, VL did not. With increasing loads, BF activity increased linearly. The authors of [23] presented a method for calculating the dynamic knee joint ROM based on the external load applied during leg extension using an EMG-driven model. According to the findings, the dynamic range of motion fluctuated depending on the external loads applied to the joint. Additionally, the authors indicated that their method will be used to design elderly rehabilitation protocols.

Finally, the goal of the study in [24] was to assess the EMG angle relationship of the quadriceps muscle during knee extensions conducted using elastic tubing and an isotonic strength training machine. Seven women and nine males were tested. EMG was gathered during the concentric and eccentric contraction phases of a knee extension, while knee joint angle was monitored using inclinometers. Experimental results indicated that there were no differences in peak EMG of the VL, VM, and RF muscles during the concentric contraction phase when comparing the machine and elastic resistance exercises.

According to the literature survey introduced above, in this paper, we propose a real-time knee extension monitoring and rehabilitation system. The aims and contributions of our system are three-fold.

First, we developed the EMG device set with two-channel surface EMG to capture EMG signals from the VL and VM muscles, and we developed an angle sensor device set using an accelerometer sensor for the knee extension ROM measurement with a range between 0 and 90 degrees. The high-performance NI-myRIO embedded device is used for central processing. Thus, both VL and VM, EMG data, and ROM data can be measured, collected, and evaluated.Second, we developed the knee rehabilitation program with the GUI using LabVIEW. Here, the rehabilitation program can be defined and flexibly set based on the user’s requirements and an expert’s recommendation. With our proposed system, the rehabilitation people can practice themselves and know their rehabilitation progress during the time of testing. Additionally, experts can review the rehabilitation progress and set the optimal treatment programs in response to users.Third, to validate and evaluate the performance of the proposed system, we perform a set of experiments on both healthy subjects and patients. Six healthy subjects whose ages ranged from 26 to 50 years and six patients aged between 44 and 79 years, including (a) knee osteoarthritis, (b) herniated disc, (c) knee ligament injury, (d) ischemic stroke, (e) hemorrhagic stroke, and (f) Parkinson are tested. Here, the patients can be divided into two groups: those with bone and joint problems and those with neurodegenerative and brain problems, which can affect the movement of the knee joint and the extensor muscles. Finally, we demonstrate bio-feedback EMG (i.e., raw data, MAV, and RMS), ROM data, and tracking results for test subjects.

According to the above objectives, the hypothesis of the study is that the knee extension monitoring and rehabilitation system can work with real-time response. It can also support users, where EMG signals and knee extension ROM data can be automatically measured and recoded based on the defined rehabilitation program.

The structure of this paper is as follows. Section 2 describes the proposed system, including the overall system, the EMG circuit and measurement, the ROM measurement, and the GUI for the rehabilitation program. Section 3 presents the details of experiments, including subject and patient information, test scenarios and rehabilitation programs, and performance evaluation indexes. Section 4 provides results and discussion conducted on healthy subjects and patients. Finally, we conclude this paper in Section 5.

## 2. Proposed System

The proposed system and the prototype developed in this work are illustrated in Figure 1 and Figure 2, respectively. The installation setup for experiments is also demonstrated in Figure 2. As in Figure 1, the proposed system consists of three major functions, including (a) two-channel surface EMG signal measurement and processing using the developed EMG device set, (b) knee ROM measurement using the accelerometer device set (or the angle sensor module in the figure), and (c) the knee rehabilitation program for users using the LabVIEW program with the GUI on the NI-myRIO embedded device (a processing unit). Note that both (a) and (b) are connected together and processed by the NI-myRIO. The NI-myRIO was used because it is a high-performance embedded device that can be efficiently applied for large data processing and also for supporting IoT and machine learning (our future work). The system can also connect to the database and Web servers through the Internet connection for monitoring and evaluation, while the physical therapists and physicians can plan the treatment using the Internet connection system [25,26]. The details of the proposed functions are described as follows.

### 2.1. EMG Measurement

To measure the EMG signals, we developed an EMG circuit (as put in the control box and connected to the NI-myRIO device in Figure 2) for two-channel EMG signal measurement of VL and VM muscles. The block diagram and the implemented circuit with the electrodes are shown in Figure 3 and Figure 4, respectively. As in Figure 3, raw EMG signals are gathered by the electrodes attached to the related muscles. These raw signals are first amplified using the instrumentation amplifier. We note that the reference drive, as connected to the reference electrode, is used as a reference for the amplifier circuit. The signals are then filtered using high-pass, low-pass, and notch filters, respectively. Here, the high-pass filter is used for gathering the high-frequency signals, where a 10 Hz cut-off frequency is applied to remove low-frequency signals caused by the effect of electrode wire movement (motion artifacts) and human effects. The low-pass filter is also applied to reduce noise at high frequencies with a cut-off frequency of 500 Hz caused by physical environments, while the 50 Hz cut-off frequency (AC base-line) can be removed by using the notch filters. With our solution and circuit development, the measured EMG signals are pre-preprocessed, and the output EMGs are the optimal signals by hardware. This can help to reduce the complexity of the preprocessing stage from a software processing perspective.

The electrodes are attached to the skin beyond the VL and VM muscles, as presented in Figure 2. Figure 5 also shows the electrode placements. The VL muscle is positioned on the thigh’s lateral side. This muscle is the largest and most strong part of the quadriceps femoris, and it serves to extend the knee joint, moving the lower leg forward. The VL muscle provides power for and absorbs the impact of activities such as walking, jumping, and running [27]. For the VM, it is an extensor muscle positioned medially in the thigh that extends the knee. The VM is part of the quadriceps muscle group. Knee pain is thought to be associated with quadriceps muscle weakness or fatigue, particularly in the VM [28,29]. Thus, both VL and VM muscles are targeted in rehabilitation because of their role in patellar stabilization during knee extension [30,31].

As demonstrated in Figure 2, two electrodes are placed on the VL and VM muscles, while one electrode is placed on the hand or forearm as the reference electrode. Finally, we also attach the 0.5, 1.0, 1.5 kg sandbag to the ankle (i.e., ankle wrist sandbag) to increase the workload of the knee during the test and also to study knee extension performances according to the rehabilitation progress. We note that we can change the weight of the sandbag based on the rehab program according to the physical therapists and physicians’ decision [33,34].

The real-time EMG signals as raw signals are also calculated in terms of mean absolute value (MAV) and root mean square value (RMS) for performance evaluation. The MAV in (1) and the RMS value in (2) of the EMG signals are also automatically calculated and reported, where x(n) is the EMG signal, and N is the number of samples to be calculated.
(1)$$MAV=\frac{1}{N}\overset{N}{\underset{n=1}{\sum }}\left|x\left(n\right)\right|$$
(2)$$RMS=\sqrt{\frac{1}{N}\overset{N}{\underset{n=1}{\sum }}{\left(x\left(n\right)\right)}^{2}}$$

### 2.2. ROM Measurement

To measure knee extension ROM, the GY-521 accelerometer sensor, as shown in Figure 6, is attached to the knee’s ankle, as demonstrated in Figure 2b. The GY-521 accelerometer is connected and communicated to the NI-myRIO embedded device through I2C, and the NI-myRIO reads accelerometer data every 100 ms. In our system, a subject is requested to sit on the chair in the posture as in Figure 6, and the subject is also requested to move the leg from 0 degree to 90 degree during the test, where 0 degree refers to the case of the subject sitting and 90 degree refers to the case of a stretched out knee. Therefore, the ROM can be measured and recorded. Here, the output of the accelerometer sensor can be transformed to θ as the degree of knee extension and using for ROM calculation. Here, $\theta_{max}$ and $\theta_{min}$ in (3) are the maximum and minimum degrees that rehabilitation people can perform (i.e., $\theta_{max}$ and $\theta_{min}$ can be varied according to the people performance), and the ROM is the absolute different between $\theta_{max}$ and $\theta_{min}$ as expressed in (4).
(3)$$\theta =\left\{\begin{array}{c} \theta_{max},\;\;\;\;If\;\theta \;is\;the\;maximum\;degree\\ \theta_{min},\;\;\;\;If\;\theta \;is\;the\;minimum\;degree \end{array}\right.$$
(4)$$ROM=\left|\theta_{max}-\theta_{min}\right|$$

The ROM value can indicate the strength of the knee. Normal people can efficiently move their knee joints for a full range of motion. On the other hand, ROM in patients, the elderly, or athletes suffering from injury, surgery, or disease, is at a low level [35,36]. Consequently, the ROM results can significantly refer to the knee’s strength, and physical therapists and physicians can review such ROM information for tracking and accurate treatment.

In our experiments, we defined that the subject begins with the starting position (0 degrees), and the subject then moves the knee to 90 degrees as the peak position (stretched out knee). For the 90-degree peak position of each subject (they can perform), we request the subject hold his or her posture for some period of time. This is called the duration of sustained contraction. Finally, the subject can return to the starting position and repeat the process again. As illustrated in Figure 7, the time before starting, the starting time, the time it takes from the initial degree to the maximum degree (i.e., peak position), the duration of sustained contraction, and the time it takes from the maximum degree to the initial degree, are demonstrated, respectively. As per our requirements, knee extension in terms of real-time ROM can be obtained and evaluated.

We note that at the beginning of our research, our ROM measurement using an accelerometer sensor was verified with the standard and professional equipment, the Con-Trex device, where experiments were performed at Songklanagarind Hospital. According to this test, the knee ROMs from the proposed device and the Con-Trex equipment could be compared, and the experimental results indicated a strong relationship between the two devices.

### 2.3. GUI for the Rehabilitation Program

The graphical user interface or GUI using the LabVIEW program from the NI-myRIO embedded device is used for setting, collecting, and presenting the patient information, the EMG (including MAV and RMS), and ROM data. Moreover, a knee rehabilitation program designed by physical therapists and physicians can be configured and assigned. Figure 8 shows the GUI monitoring window. There are three major parts as in Figure 9, including (a) patient information, (b) rehabilitation, and (c) results, respectively.

**Patient information**: Patient information such as patient ID, name, age, gender, and the date and time of treatment sessions can be recorded. Target value information, including MAV, RMS, and ROM thresholds, is also assigned for patients.**Rehabilitation**: The therapy training at present can be set. During the test, real-time EMG and ROM signals measured can be automatically displayed and recorded. We include start and stop buttons for training control in this rehabilitation window. We intend to immediately stop the training program during the test if the patient is unable to perform their action due to a medical condition. This can help to achieve the patient’s safety.**Results**: The average MAV, RMS, and ROM data calculated from each time window of each obtained signal are calculated and displayed. As seen in Figure 8c, MAV1 and RMS1 are for the VL muscles, while MAV2 and RMS2 are for the VM muscles. All EMG and ROM signals are displayed, and all measured data can be recorded to the database using the upload button. Finally, output results, which are calculated by comparing the obtained MAV, RMS, and ROM values with the target values (i.e., MAV, RMS, and ROM thresholds), are presented. Here, 1 refers to passing and 0 refers to not passing. Thus, physical therapists and physicians can suddenly review and evaluate rehabilitation results. We note that, by this approach, rehabilitation people who perform their training can also know their rehabilitation progress during the time of testing.

Figure 9 provides the summary of the work flow of the knee extension monitoring and rehabilitation system. The program starts with entering patient data into the system to retrieve any related data from the database. Then, the correct installation of the EMG electrodes and the angle sensor for ROM are checked by observing the signal displayed on the monitoring window. The number of repetitions of the physical therapy for each test, the number of tests, and the knee side to be tested are assigned. Here, the program immediately starts when the patient is ready for testing. The program automatically counts and checks the number of repetitions of the physical therapy. If the count is not reached, the patient is allowed to continue the action until the number of repetitions has been completed. Therefore, if yes, the EMG and ROM data of such a test are automatically recorded. Finally, when the number of tests reaches the defined level, all measured data are recorded and processed and the program stops when it completes the rehabilitation program. We note that the specification information of our proposed device and system is also provided in Table 2.

## 3. Experiments

### 3.1. Subject and Patient Information

The proposed system is verified and tested by six healthy subjects and six patients who receive treatment at a physical therapy unit in Trang Hospital, Thailand. The subject information, including gender, age, weight, the effected knee, and related important information, are provided in Table 3a,b, respectively.

For the healthy subjects as in Figure 10, they are both male and female with an age between 26 and 50 years. Their body size and the effected knee are also different. Additional information about exercise behavior is also provided. Note that, for the 5th subject, he always exercises (playing football) 3–5 times a week, and he had knee surgery on his left knee due to hard exercise. An illustration of his knee muscles is also shown in Figure 11, where the VL and VM muscle sizes on his left side are smaller than those on his right side.

For the patients, as illustrated in Figure 12, the subjects are divided into two groups. The first group is composed of patients with bone and joint problems, including (a) knee osteoarthritis, (b) herniated disc, and (c) knee ligament injury. The second group is the patients with neurodegenerative and brain problems, including (a) ischemic stroke, (b) hemorrhagic strokes, and (c) patients with Parkinson’s disease. Here, these kinds of diseases can affect the movement of the knee joint and the extensor muscles.

For the patient selection criteria based on physical therapist and physician’s decision, inclusion criteria includes (a) patients over 18 years old, (b) knee extension movement higher than 10 degrees, (c) patients with extension movement problems, and (d) patients who agree and sign the consent form for participating in the research study. Exclusion criteria includes: (a) patients who are unable to move their knee; (b) patients who have a lesion in the knee joint area; (c) patients who have inflammation and swelling in the knee joint area; and (d) patients who have impaired knee flexion. For subject withdrawal criteria, there are (a) patients who request to withdraw from the research study and (b) patients who are in the research study but they are considered to be in the group of exclusion criteria during the study. The inclusion criteria, exclusion criteria, and the subject withdrawal criteria are also summarized in Table 3c.

### 3.2. Test Scenarios and Rehabilitation Program

The healthy subjects are tested on one day, while for the patients, they are tested three times on different days to track their knee rehabilitation progress. Here, the patients are tested on one month, where each test has the same duration (first week, middle week, and last week of the month), as assigned by physical therapists and physicians. Note that before the first test, patients who received treatment at the physical therapy unit, department of rehabilitation at Trang Hospital had been continuously practicing for the rehabilitation under the supervision of experts. However, there is no measurement device, like our proposed device for rehabilitation tracking.

All tests are performed in a sitting position. Before the test, the subject’s skin beyond the VL and VM muscles is cleaned with an alcohol pad. The locations of the electrodes are marked to ensure the same measurement area for other tests. In this work, physical therapists and physicians are asked to mark and set the optimal positions for surface EMG measurement. Testing of the connection is also rechecked and monitored by experts to obtain the correct surface EMG signals. Finally, before, during, and after the test, electrodes and signals are rechecked both in terms of physical material, usability, and signal quality. The angle sensor is attached around the knee’s ankle. The subject begins with the starting position at 0 degrees, where 0 degrees refers to the case of sitting. We note that for the healthy subjects, we attach the 1.0 kg sandbag to their ankles to increase the workload of the knee during the test. For the patients, the weight of the sandbag is selected according to the expert’s decision.

The rehabilitation program set for both healthy subjects and patients is demonstrated in Figure 13. In each test, each subject has five rounds for his or her knee movements. In each round, the subject has to move his knee angle from the starting position (or 0 degrees) to the peak position (or 90 degrees if they can). The subject is recommended to hold his knee at the peak position for 3 to 6 s, approximately, before returning back to the starting position. After the first round is done, the subject rests for a minute, and then the second round is started. The program stops when the five rounds (or 1 set) are reached and the EMG signals and knee extension ROM data are completely recoded and processed.

### 3.3. Performance Evaluation Indexes

To investigate the performance of the proposed system, the real-time EMG signals, including the raw signals from VL and VM muscles, are measured. The MAV in (1) and RMS in (2) of the EMG signals for both VL and VM are also automatically calculated. For knee extension ROM data, the real-time knee angle in degrees is measured, and the average ROM is also reported. Additionally, we also provide rehabilitation progress results for patients.

## 4. Results and Discussion

### 4.1. Healthy Subjects

The real-time ROM results and the EMG signals of VL and VM muscles measured from the healthy subjects are demonstrated in Figure 14 and Figure 15, respectively. Table 4 and Figure 16 also show the average MAV and the RMS of the EMG signals for both VL and VM and the average ROM, and Figure 17 also illustrates an example of experimental results displayed in the GUI window.

The results demonstrate that the proposed system can successfully measure the real-time ROM and EMG signals according to the rehabilitation program performed by the subjects. The results also indicate that when the subjects begin to move their knee joints against the weight of the sandbag that affects VL and VM muscles, the EMG signal amplitude then increases and the ROM can be read according to the knee movement degree. Here, the VL and VM EMG signals have the maximum amplitude when the subjects move their knee to their peak position. The EMG signal amplitude gradually decreases when the subjects hold their knee at the peak position for a period of time due to the fatigue of their muscles (for some subjects). Finally, the EMG signal amplitude then significantly decreases to the lowest level when the subjects return to the starting position. As seen in Figure 14 and Figure 15, the ROM results also correlate with the EMG results, where the subjects have a ROM movement degree of 0 to 80, or nearly 90 degrees, approximately. We note that, from our implementation, Figure 17 also confirms that the ROM and the EMG can be collected and displayed at the same time.

The results from Table 4 and Figure 16 also reveal that the amplitude of the VL EMG signal of each subject is larger than the case of the VM EMG signal. Here, this is because the subject put more force from the VL muscle against the sandbag weight than the VM muscle. From Figure 15 and Figure 16, we can also observe that the first subject who always exercises 3–5 times a week has high EMG signal amplitude and smoothed ROM results. Both VL and VM signal amplitudes are stable when he holds his knee at the peak position for a period of time. Also, the ROM is smooth for five rounds of his tests. The 6th subject, who always exercises, also performs well, as indicated by her EMG and ROM results. For the 4th subject, since he has a quite high weight of 110 kg, his EMG signals captured from his muscles are not very big, like in the cases of the 1st and 6th subjects. Finally, for the 5th subject, although he had knee surgery on his left knee, he had passed the rehabilitation process and he always exercises 3–5 times a week. Thus, he obtains good EMG signals and ROM results also. However, when compared to the first subject, the VL and VM EMG signal amplitudes are lower. We note that, as shown in Figure 11, the VM muscle of the knee left side of the 5th subject is significantly smaller than the VM muscle of his knee right side.

### 4.2. Patients

The real-time ROM and the EMG signals (from the first week and the last week of testing) obtained from six patients, including (a) knee osteoarthritis, (b) herniated disc, (c) knee ligament injury, (d) ischemic stroke, (e) hemorrhagic stroke, and (f) Parkinson, are presented in Figure 18, respectively. Table 5 and Table 6 also provide the summary of the average ROM, the MAV, and the RMS of the EMG signals. The experimental results from our test reveal that the patients with bone and joint problems have a greater capability of knee movement than the patients with neurodegenerative and brain problems due to bone and joint problems with muscle strength and a good commanding system. For the 1st to 3rd subjects, the ROM can reach its maximum peak value and it is smooth for all five rounds of their tests, while the EMG signals have high amplitude for both VL and VM amplitudes. However, compared with the healthy subjects such as the first subject as in Figure 14 and Figure 15, the EMG signals from the patients are not stronger since both the VL and VM signal amplitudes of the first healthy subject are more stable when he holds his knee at the peak position for a period of time. For patients with neurodegenerative and brain problems, we can observe that their ROM results are not stable in terms of ROM peak values and a period of time to hold their knee. Additionally, the VM and VL EMG signal amplitudes are quite small compared with the cases of the patients with bone and joint problems.

As we mentioned in Section 3, the patients who received treatment at the rehabilitation department at Trang Hospital in Thailand were tested three times on different days to track their knee rehabilitation progress. They were tested for one month, i.e., the first week, the middle week, and the last week of the month. Before the first test, patients had been continuously practicing for the rehabilitation under the supervision of physical therapists and physicians, Table 6. Table 5 and Table 6 and Figure 18 show the results obtained from the first week and the last week of the patients’ rehabilitation, where the average ROM, the MAV, and the RMS of both VL and VM EMG signals are demonstrated. For the patients in group A, the average ROMs are 85.5 (first week) and 87.167 (last week), while they are 78.7 and 81.467 for the patients in group B. In cases of EMG results as presented by the MAV and the RMS values, on average, they are different from the first week to the last week for both VL and VM muscles, as summarized in Table 6 and also in Table 5. For the second patient with a herniated disc, almost all EMG results are increased, but the MAV and the RMS of the VM muscle of the first week and the last week are not different (MAV: 0.068; first week, 0.067; last week) and (RMS: 0.094, 0.096), while the MAV and the RMS of the VL are increased. Thus, experts can see these results and give more consideration to this case.

The comparison of the average ROM, the MAV and the RMS of the EMG signals between the healthy subject group and the patient group is also illustrated in Table 7 and Figure 19. The results here demonstrate that we can monitor and track the rehabilitation progress of the patients compared with themselves and the healthy subject group. As in Table 7, the ROM averaged from all patients is 82.1 for the first week and then moves to 84.317 for the last week. However, if we consider the healthy subjects’ information with the ROM of 87.73, the patient group can still practice themselves to increase to a higher level of the ROM, as much as possible. For the EMG results of the patients, both MAV and RMS of the VL and VM muscles can be improved by rehabilitation. Figure 19 shows the ROM signal slope obtained from the 1st healthy subject and the 4th patient with ischemic stroke (first week). Here, the signal slope of the 4th patient is not strong during the duration of sustained contraction (holding his posture at the peak position for some period of time), compared to the 1st subject. However, after rehabilitation, the ROM of the 4th patient improved, as seen in Figure 18d. Moreover, the MAV and RMS of the EMG signals are all improved as well.

## 5. Conclusions

A real-time knee extension monitoring and rehabilitation system is proposed in this paper. The proposed system consists of an EMG device with two-channel surface EMG to capture EMG signals from the VL and VM muscles and an angle sensor device for the knee extension ROM measurement. In addition, the knee rehabilitation program with the GUI using LabVIEW on the NI-myRIO embedded device is also provided, where the knee rehabilitation program can be defined and flexibly set, as suggested by physical therapists and physicians. The proposed system is tested by both healthy subjects and patients. Here, patients with bone and joint problems as well as those with neurodegenerative and brain problems are included in the study. The experimental results demonstrate real-time bio-feedback EMG (i.e., raw data, MAV, and RMS) and ROM data. Additionally, the tracking results of the rehabilitation progress of the patients are also provided. With our system and methodology, the rehabilitation people can know their progress during the testing, while the experts can check the progress and set the optimal treatment programs in response to the rehabilitation people.

Limitations of this study can be summarized in three points. First, since the surface EMG device and the angle sensor device are placed at different positions and they are connected to the NI-myRIO via wire communication to increase usability and convenience, they should be replaced by a wireless connection. Second, to more accurately use the surface EMG electrodes, they should be placed on the correct positions of the VL and VM muscles. In this work, physical therapists and physicians are asked to mark and set the optimal positions. Therefore, if patients use the device by themselves, they should be appropriately trained first. Finally, the results obtained from this work are the real-time EMG with MAV and RMS and the ROM data. Extracted information from such data for greater utilization should be provided; for example, the ROM variation, the time period the user can hold his or her knee above the threshold level, etc.

In future work, more investigation and evaluation of the relationship between ROM and VL/VM EMG measured from a large number of healthy subjects and patients will be focused. An interval time to reach the maximum peak will be calculated for evaluation. The designed protocol related to this issue will be implemented in our proposed system for testing and evaluation. Additionally, the feature extraction and the classification of knee extension problems and rehabilitation progress by machine learning will be taken into consideration. Finally, the rehabilitation progress of the patients according to the desired treatment programs assigned by physical therapists and physicians will be followed up and evaluated.

## Figures and Tables

**Figure 1 healthcare-10-02544-f001:**
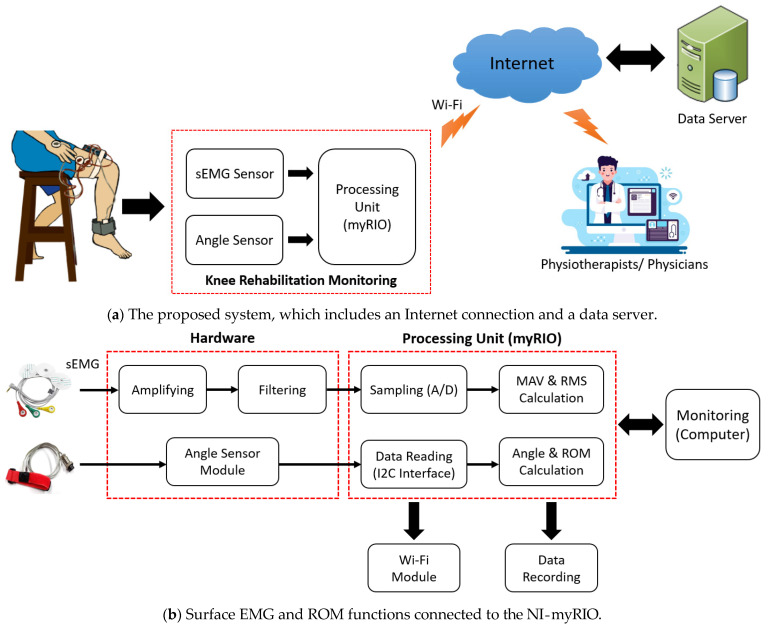
Knee extension monitoring and rehabilitation systems.

**Figure 2 healthcare-10-02544-f002:**
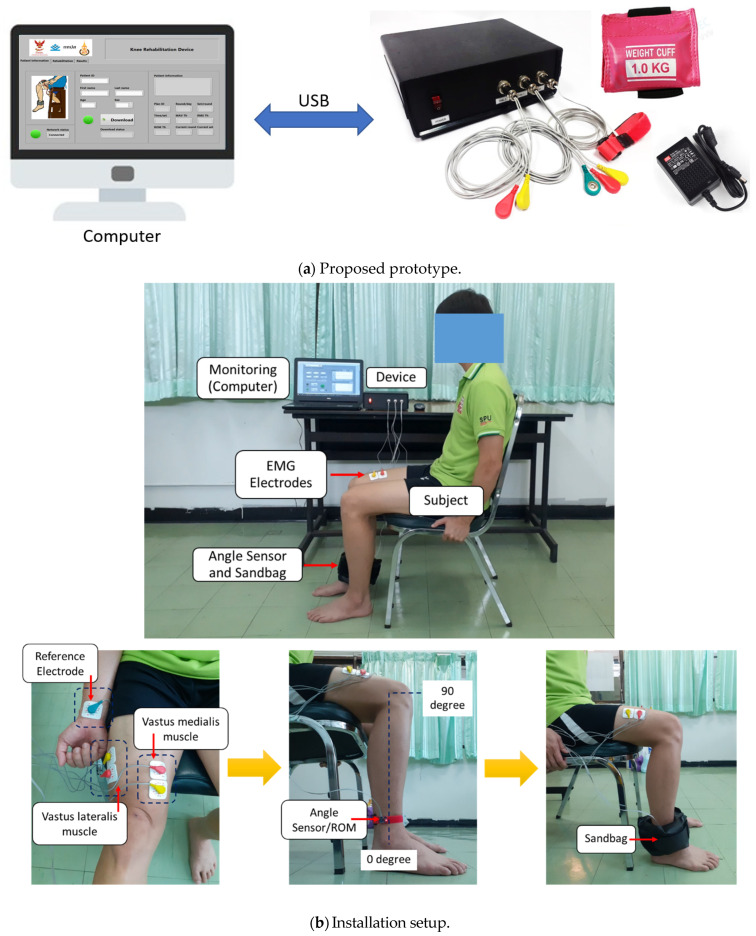
The prototype and installation setup.

**Figure 3 healthcare-10-02544-f003:**
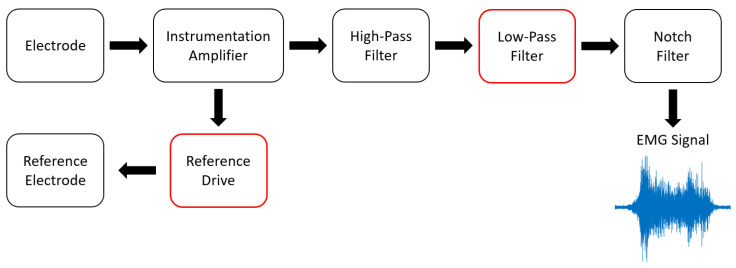
A block diagram of the EMG measurement circuit.

**Figure 4 healthcare-10-02544-f004:**
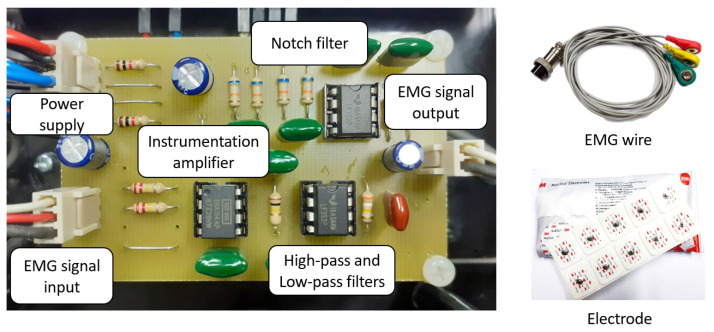
EMG measurement circuit, EMG wires, and electrodes.

**Figure 5 healthcare-10-02544-f005:**
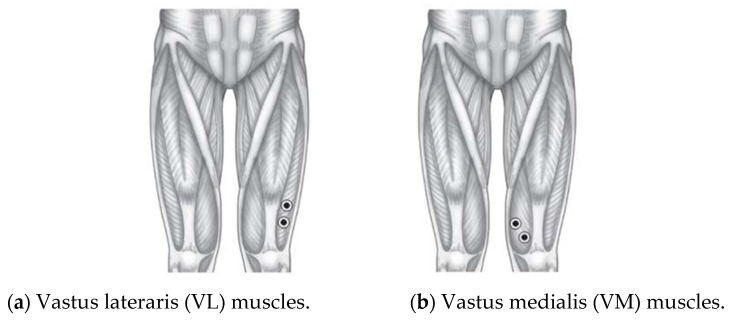
Positions of the electrodes [32].

**Figure 6 healthcare-10-02544-f006:**
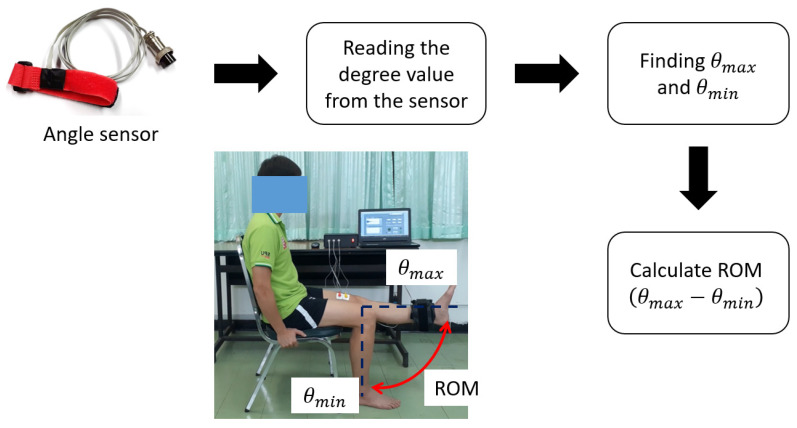
ROM calculation: angle sensor and ROM of knee extension (0 to 90 degrees).

**Figure 7 healthcare-10-02544-f007:**
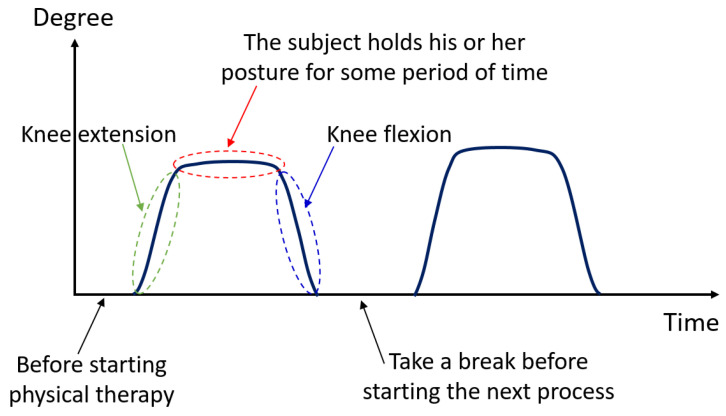
An example of the ROM pattern with a 0 degree to 90 degree designed for testing.

**Figure 8 healthcare-10-02544-f008:**
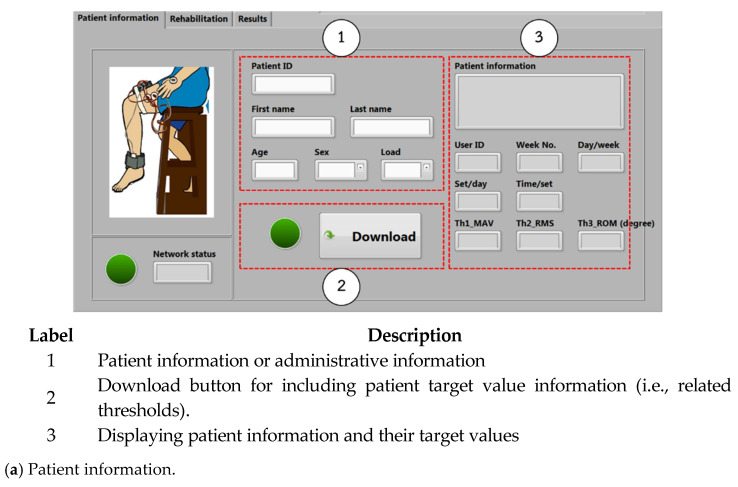
GUI monitoring window.

**Figure 9 healthcare-10-02544-f009:**
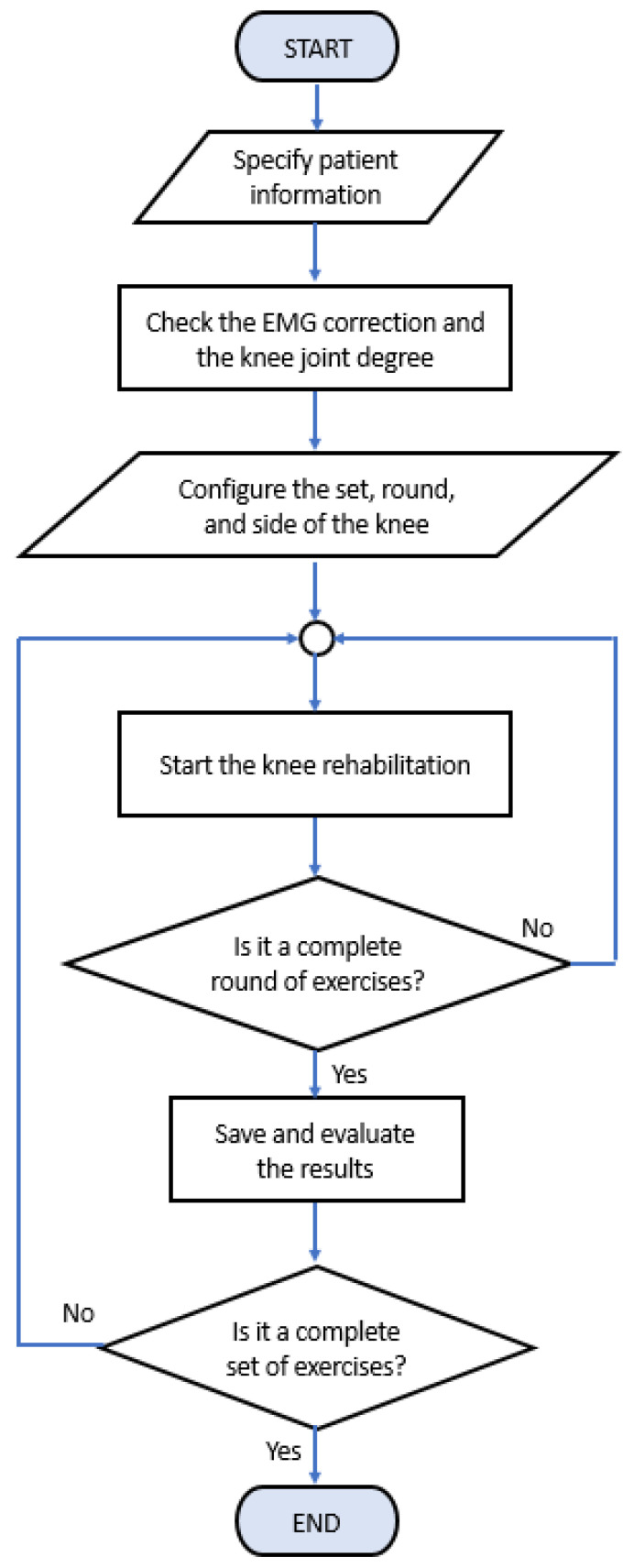
The work flow of the knee extension monitoring and rehabilitation system.

**Figure 10 healthcare-10-02544-f010:**
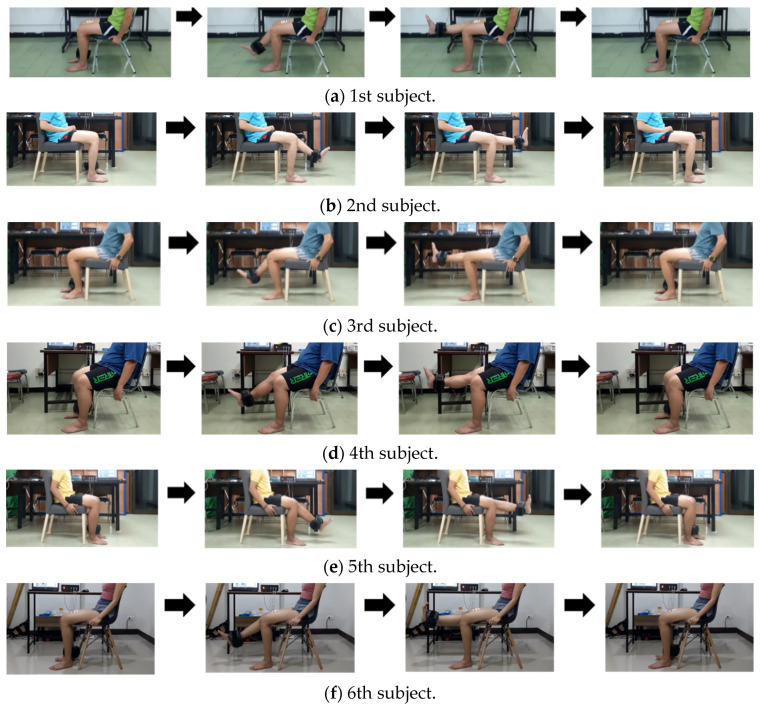
The test of six healthy subjects.

**Figure 11 healthcare-10-02544-f011:**
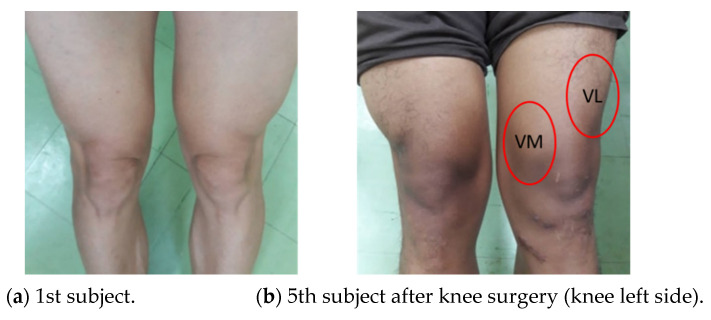
An illustration of the knee muscles of the 1st subject and the 5th subject who had knee surgery on their left side.

**Figure 12 healthcare-10-02544-f012:**
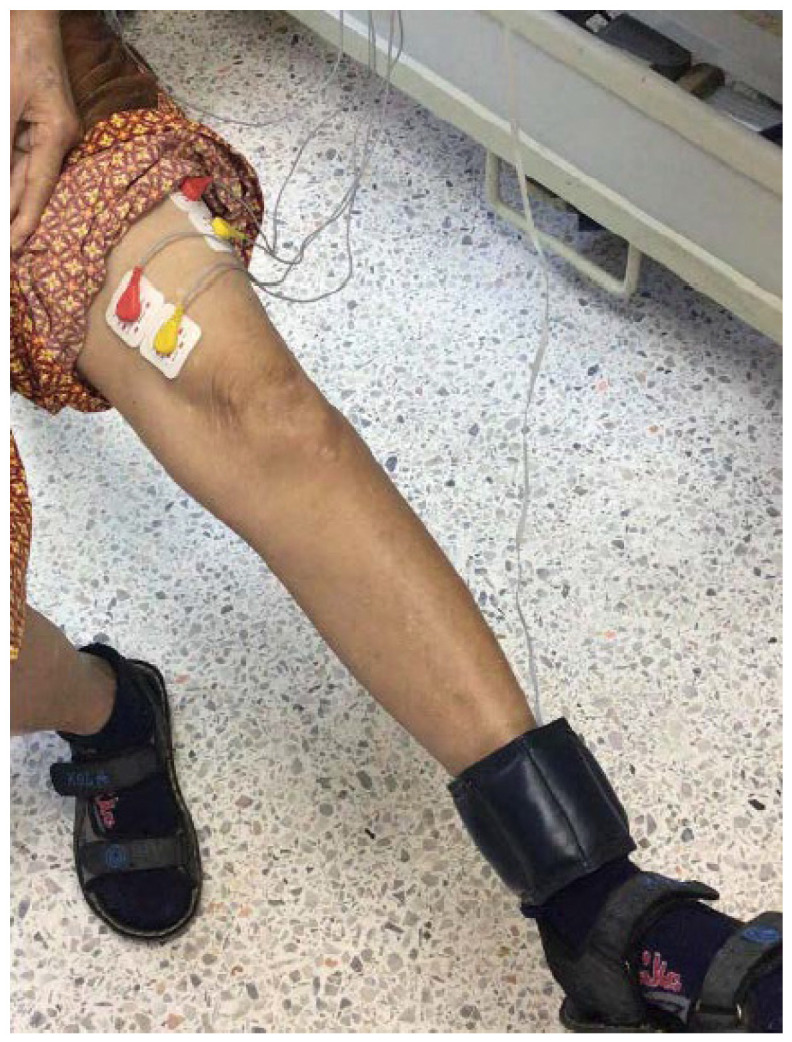
An illustration of the test of the patient.

**Figure 13 healthcare-10-02544-f013:**
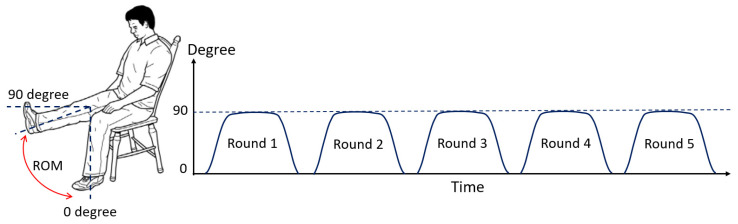
Rehabilitation program for healthy subjects and patients.

**Figure 14 healthcare-10-02544-f014:**
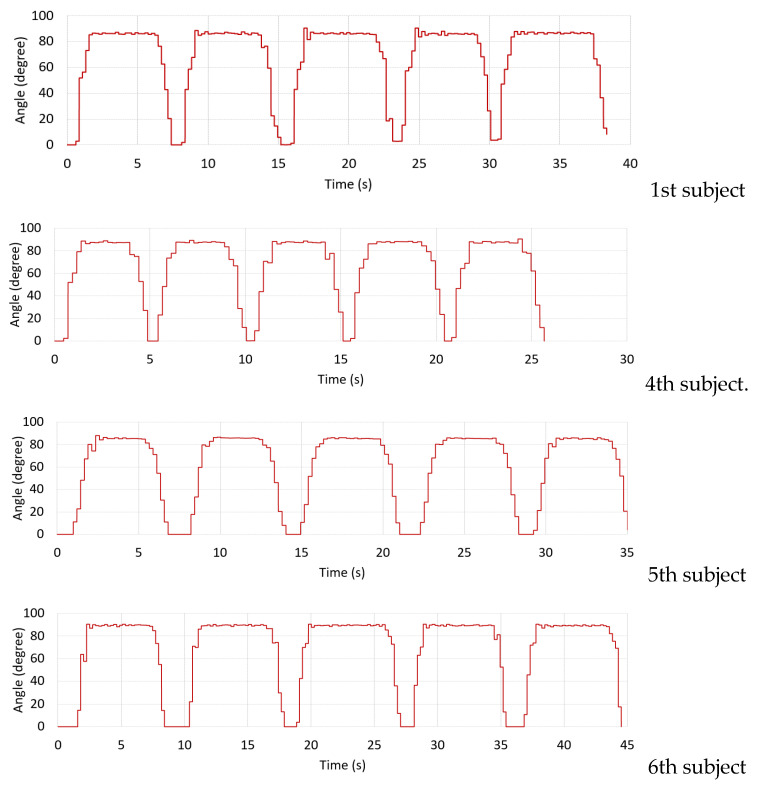
ROM results of the healthy subjects.

**Figure 15 healthcare-10-02544-f015:**
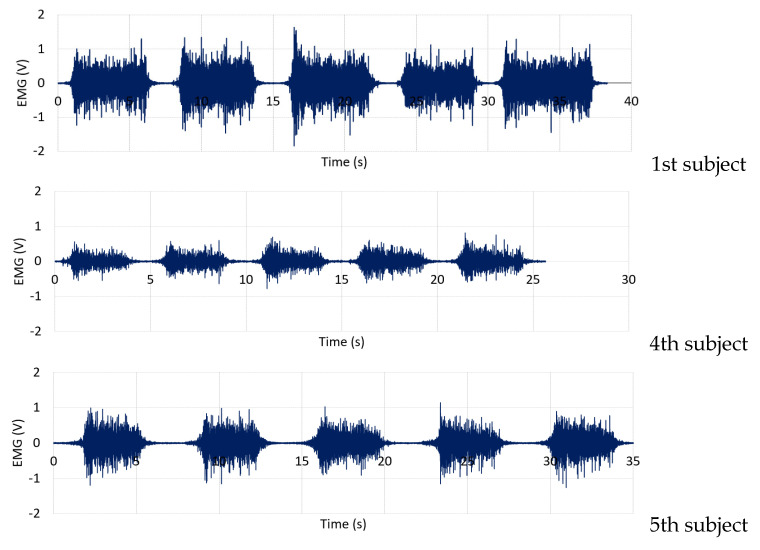
EMG signals from the VL and the VM muscles of the healthy subjects.

**Figure 16 healthcare-10-02544-f016:**
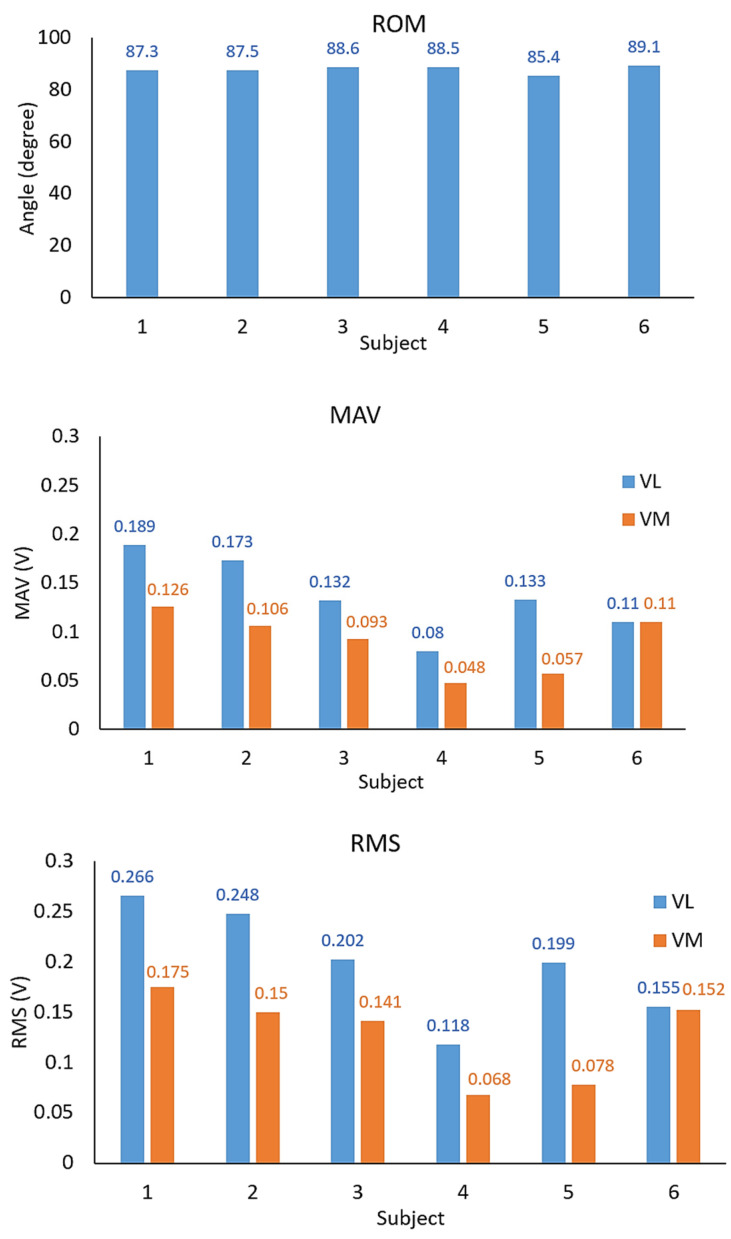
Summary of the average ROM, the MAV, and the RMS of the EMG signals.

**Figure 17 healthcare-10-02544-f017:**
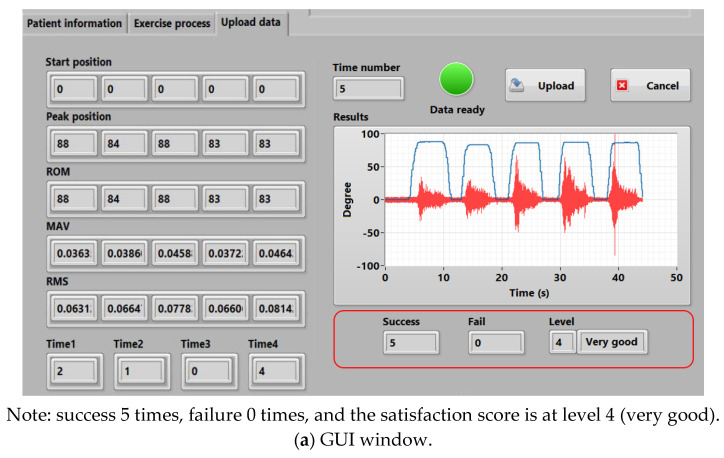
An example of experimental results displayed in the GUI window.

**Figure 18 healthcare-10-02544-f018:**
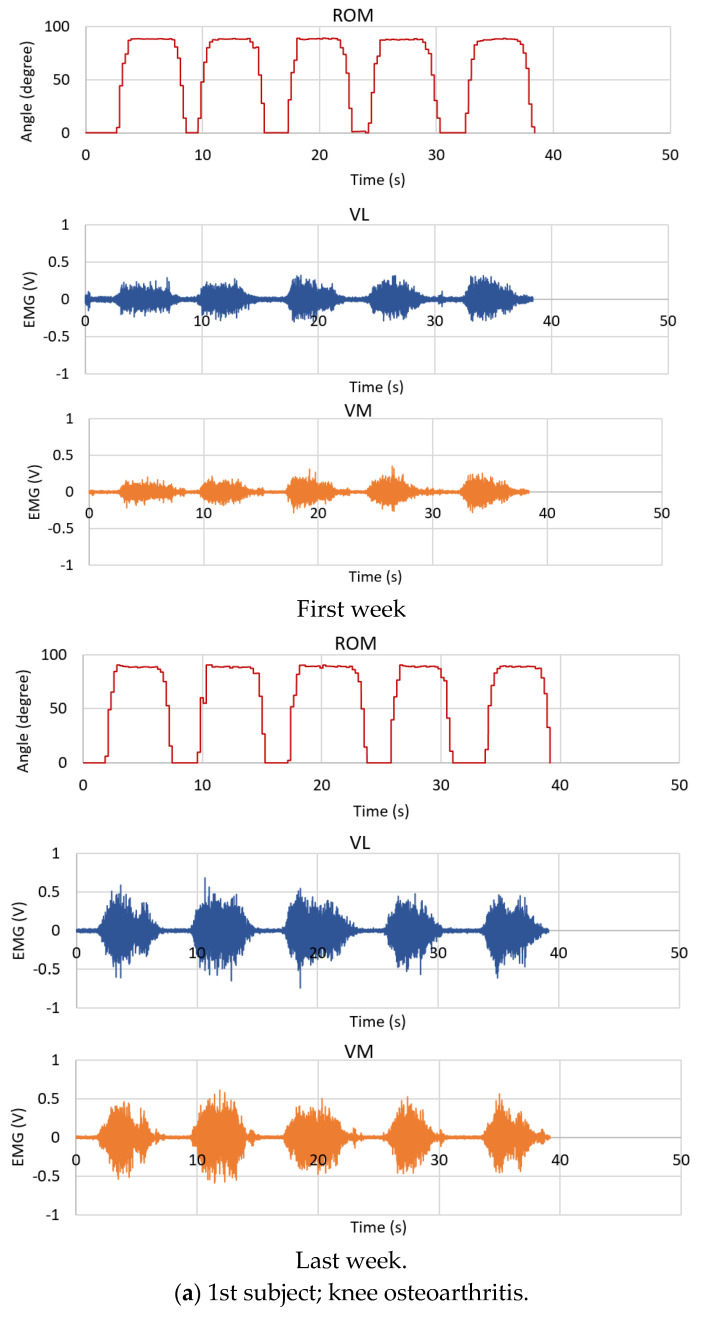
ROM and EMG signals obtained from the patients.

**Figure 19 healthcare-10-02544-f019:**
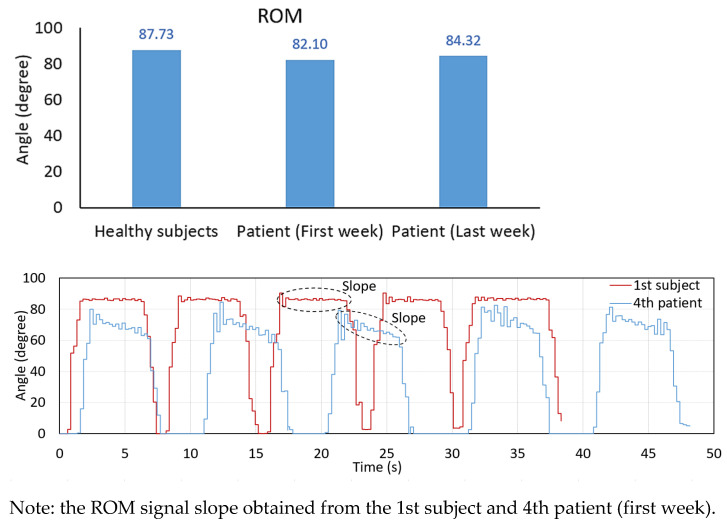
Comparison of the ROM, the MAV, and the RMS of the EMG signals between the healthy subjects and the patients.

**Table 1 healthcare-10-02544-t001:** A summary comparison between related works and this work.

Ref.	Objectives	Technologies/Sensors
**Video camera-based**:		
[9]	Human motion analysis Lower limb kinematics were measured.	Kinect
[10]	Stroke rehabilitation A rehabilitation system to detect the gestures of patients when lying in bed	3D camera/Intel RealSense D415
[11]	The mobility of patients (i.e., the Timed-Up-and-Go (TUG) test) was observed.	Computer vision with a regular RGB video camera
**Wearable sensors**:		
[12]	To assess motor performance in post-acute stroke patients	3-axis gyroscope; 3-axis accelerometer; 3-axis magnetometer; Barometer
**ROM**:		
[13]	A sensor fusion technique was developed to predict lower-limb joint angles (knee joint angle) in real-time.	Accelerometer; Gyroscope
[14]	A sensory motion system that estimates angular variations in the knee and ankle joints was developed to diagnose gait problems.	Accelerometer; Gyroscopes; Magnetometer
[15]	To assess ROM for robot-assisted ankle-foot rehabilitation.	A robotic ankle–foot rehabilitation system
[16]	How sensor position affects the precision of knee ROM information gathered by a rehabilitation system was studied.	The BPMpathway rehabilitation system A BIOPAC twin-axis digital goniometer
**EMG**:		
[17]	The influence of shoe heel height on VL and VM EMG activity during sit-to-stand transitions was investigated.	EMG sensor
[18]	A home-use EMG-controlled knee exoskeleton to assist stroke patients in their rehabilitation was demonstrated.	Myo with eight active channels EMG sensor (Ch1 to Ch8)
[19]	The Arduino Mega board and an EMG muscle sensor were used to develop a low-cost EMG system. EMGs from the rectus femoris muscle were recorded while performing isometric and dynamic activities.	EMG muscle sensor
[20]	A rehabilitation approach based on a lower-limb exoskeleton integrated with a human-machine interface was introduced EEG and multichannel EMG data from leg muscles were recorded.	EEG and multichannel EMG sensors
**ROM and EMG**:		
[21]	An isokinetic machine (i.e., the Biodex) was used to measure knee joint resistive torque during passive motion.	Isokinetic machine (i.e., the Biodex) EMG electrodes
[22]	The association between isometric force and surface EMG of quadriceps femoris muscles in the single-joint knee extension and multi-joint leg press exercises was investigated. EMGs from the VL, VM, RF, and BF were recorded.	Dynamometer; EMG electrodes
[23]	An EMG-driven model is used to calculate the dynamic knee joint ROM based on the external load delivered during leg extension exercise.	Dynamometer EMG electrodes
[24]	To assess the EMG angle relationship of the quadriceps muscle during knee-extensions	An isotonic strength training machine and elastic tubing; EMG electrodes
**This work**	A real-time and autonomous knee extension monitoring and rehabilitation system was developed and tested. ROM and EMG were measured for evaluation. A knee rehabilitation program could be defined and flexibly set for rehabilitation training. Both healthy subjects and patients related to knee extension rehabilitation were tested and evaluated.	Angle sensor; Surface EMG electrodes; NI-myRIO embedded device; GUI and the rehabilitation program using LabVIEW

**Table 2 healthcare-10-02544-t002:** Specifications of the proposed device and system.

Physical Specification (Control box)	
Application	Operating room
Operating temperature	0–40 °C
Dimension (W × L × H)	15 cm × 20 cm × 8 cm
**Control Unit Specification (NI** **-miRIO)**	
Processor	Xilinx Z-7010
Supply voltage	12 Vdc
Power consumption	2.6 W
ADC resolution	12 bit
ADC sample rate (max)	500 kS/s
**Performance Specification**	
EMG channel	2 channels for EMG signal measurement
Angle range	0–90 degrees for ROM measurement
Data collection/ Knee rehabilitation program	- Support multiple users - Autonomous data collection and processing - Rehabilitation programs can be defined and flexibly set for rehabilitation users, as recommended by experts. - Experts can review the rehabilitation progress and set the optimal treatment programs in response to the users.

**Table 3 healthcare-10-02544-t003:** Subject and patient information.

(a) Healthy subjects.					
**Subject**	**Gender**	**Age** (**Year**)	**Weight** (**Kg**)	**Knee** **’** **s Side**	**Note**
1	Male	35	65	Right	Always exercise 3–5 times a week
2	Male	42	72	Left	-
3	Male	50	76	Right	-
4	Male	26	110	Right	-
5	Male	31	67	LeftAfter knee surgery	Always exercise 3–5 times a week
6	Female	30	49	Right	Always exercise 3–5 times a week
(b) Patients.					
**Subject**	**Gender**	**Age** (**year**)	**Weight** (**kg**)	**Knee** **’** **s Side**	**Note**
Group A: Patients with bone and joint problems					
1	Female	52	86	Left	Knee osteoarthritis
2	Female	68	59	Left	Herniated disc
3	Female	44	52	Right	Knee ligament injury
Group B: Patients with neurodegenerative and brain problems					
4	Male	77	48	Right	Ischemic stroke
5	Male	46	68	Left	Hemorrhagic stroke
6	Female	79	57	Left	Parkinson
(c) The inclusion criteria, exclusion criteria, and subject withdrawal criteria.					
**Inclusion criteria**		**Exclusion criteria**		**Subject withdrawal criteria**	
Patients with an age above 18 years old Knee extension movement higher than 10 degree Patients extension movement problems Patients who agree and sign the consent form to participate in the research study		Patients who cannot move their knees Patients who have a lesion in their knee area Patients with inflammation and swelling at the knee joint area Patients with impaired knee flexion		Patients who request to withdraw from the research study Patients who are in the research study but they are considered to be in the group of excluded criteria during the study.	

**Table 4 healthcare-10-02544-t004:** Summary of the average ROM, the MAV, and the RMS of the EMG signals.

Healthy Subject	ROM (Degree)	MAV (V)		RMS (V)	
	Average Value	VL	VM	VL	VM
1	87.3	0.189	0.126	0.266	0.175
2	87.5	0.173	0.106	0.248	0.15
3	88.6	0.132	0.093	0.202	0.141
4	88.5	0.08	0.048	0.118	0.068
5	85.4	0.133	0.057	0.199	0.078
6	89.1	0.11	0.11	0.155	0.152

Note: the average result of the signals from the five sets of testing.

**Table 5 healthcare-10-02544-t005:** Summary of the average ROM, the MAV, and the RMS of the EMG signals.

(**a**) **First week**.					
**Patient**	**ROM (** **Degree)**	**MAV (** **V)**		**RMS (** **V)**	
	**Average Value**	**VL**	**VM**	**VL**	**VM**
1	87.5	0.046	0.036	0.063	0.05
2	80.1	0.083	0.068	0.12	0.094
3	88.9	0.088	0.104	0.127	0.152
4	74.2	0.025	0.022	0.044	0.04
5	78.6	0.043	0.036	0.062	0.05
6	83.3	0.043	0.027	0.063	0.037
(**b**) **Last week**.					
**Patient**	**ROM (** **Degree)**	**MAV (** **V)**		**RMS (** **V)**	
	**Average Value**	**VL**	**VM**	**VL**	**VM**
1	88.4	0.083	0.078	0.117	0.112
2	85.1	0.091	0.067	0.134	0.096
3	88	0.1	0.111	0.148	0.163
4	77.6	0.04	0.033	0.063	0.049
5	80.1	0.073	0.041	0.11	0.059
6	86.7	0.082	0.039	0.119	0.052

**Table 6 healthcare-10-02544-t006:** The average ROM, the MAV, and the RMS of the EMG signals; classification by group.

	ROM (Degree)	MAV (V)		RMS (V)	
	Average Value	VL	VM	VL	VM
First week					
Patient (Group A)	85.5	0.072333	0.069333	0.103333	0.098667
Patient (Group B)	78.7	0.037	0.028333	0.056333	0.042333
Last week					
Patient (Group A)	87.16667	0.091333	0.085333	0.133	0.123667
Patient (Group B)	81.46667	0.065	0.037667	0.097333	0.053333

Note: Group A: patient 1, 2, and 3 with bone and joint problems. Group B: patient 4, 5, and 6 with neurodegenerative and brain problems.

**Table 7 healthcare-10-02544-t007:** Comparison of the average ROM, the MAV, and the RMS of the EMG signals between the healthy subjects and the patients.

	ROM (Degree)	MAV (V)		RMS (V)	
	Average Value	VL	VM	VL	VM
Healthy subjects	87.73333	0.136167	0.09	0.198	0.127333
Patients (First week)	82.1	0.054667	0.048833	0.079833	0.0705
Patients (Last week)	84.31667	0.078167	0.0615	0.115167	0.0885

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
