# Peer review of "Development of a Real-Time Knee Extension Monitoring and Rehabilitation System: Range of Motion and Surface EMG Measurement and Evaluation"

_healthcare, 2022, doi:10.3390/healthcare10122544_

Round 1

Reviewer 1 Report

Dear Authors,

It is my pleasure to review your study but I have a few doubts.

-First of all, the article is too long, too extensive, contains too much data, is interesting but very difficult to read. I think it needs improvement.

-It is very extensive. It is worth focusing on key information.

-In the line 74: "The most important treatments for osteoarthritis are rehabilitation and exercise to strengthen the knee muscles."

I can't agree with the authors. Treatment of OA is multidisciplinary and depends on the stage of degenerative changes.

-No raim of study. No hypothesis. This should be corrected.

-Looking at such a long text there should be more references. It's worth adding more references, especially new ones.

-No limitations of study. This should be corrected.

-The study is based on a small number of patients. 

-The graphics have been well prepared and are of good quality.

Author Response

Dear Editors and Reviewers,

We are grateful for your comments on our manuscript.

We revised the manuscript according to your advice.

Below is a point-by-point response to your comments.

Reviewer 2 Report

Manuscript ID: healthcare-2071842

Manuscript title: Development of a Real-Time Knee Extension Monitoring and Rehabilitation System: Range of Motion and Surface EMG Measurement and Evaluation

Comments

The manuscript reports a study proposing a real-time knee extension monitoring and rehabilitation system. The system is based on a two-channel surface electromyography and angle sensor with a software-based graphical interface developed in LabVIEW. The system is tested in 12 participants (6 healthy people and 6 people with neurologic and/or musculoskeletal conditions).

The background is a little long but seems comprehensive for the field. The table helps summarize the main background topics. The study aims are clear and is adequate to the study design. 

Major comments

1. Methods. Consider visiting the EQUATOR Network to seek appropriate guidelines for reporting case series (e.g., https://www.equator-network.org/reporting-guidelines/care/).

2. Line 256. I missed some information about the guidelines for surface electromyography. For instance, the SENIAM project (http://www.seniam.org) and ISEK (https://isek.org/emg-standards) suggest technical and methodological aspects that must be observed to obtain high-quality measurements. For instance, there are several specifications for the electronic device that have minimal requirements and are not reported for this new device. 

3. Lines 333-334. What is the accuracy of the range of motion as per your protocol? How can you ensure a participant obtains exactly 90 degrees at the peak position?

4. Table 5 and Figure 19 show the same result. Given the case series design, I suggest keeping only Table 5.

5. Line 693 (and other text passages). To report that patients ‘improved’ between the first and last weeks of interventions you need either to report confidence intervals (preferable) or a null hypothesis significance test (not recommended given the small sample and study sampling).

6. Lines 723.724. Please consider toning down the validation aspect as a two-group case series in a quasi-experimental (pre vs. post) study design is not sufficient to show the validity of new devices.

7. I missed some discussion about the limitations and strengths of the study for a more comprehensive assessment of your findings.

8. References seem mostly adequate, but consider revisiting them for more updated sources.

Minor comments

1. Rehabilitation is a much broad, multi-professional intervention than that defined in the manuscript. Please report more particularly what intervention was actually delivered.

Author Response

(The authors gave the same response as above.)

Reviewer 3 Report

According to the content of the paper, both EMG data and ROM data can be measured by the proposed system. Additionally, the rehabilitation program can be defined based on the user’s requirements. A set of experiments were performed to validate and evaluate the performance of the system. But would it be more convincing if the sample size was increased? Moreover, readers may want to know the shortcomings of the system and the limitations of the article. 

Author Response

(The authors gave the same response as above.)

Reviewer 4 Report

This is a very well written paper as it describes the procedures and methods of this study in detail and provides some very interesting results. I also think that the paper provides information on a great system.

On the other hand, I think the background should be more concise. Also, you have provided big data and figures and more information, but I would suggest that you reconsider the way you provide the figures. 

For example, you provide figures of ROM, EMG, etc. for each subject, but I feel that it is difficult to understand individual characteristics from figures.

I have no specific comments, but please consider if necessary how to simplify the paper as a whole and how to provide figures and data.

Author Response

(The authors gave the same response as above.)

Round 2

Reviewer 1 Report

Dear Authors,

the article looks much better.

Thanks for making the corrections.

It is still too long and therefore hard to read but I think manuscript can be published in Healthcare (ISSN 2227-9032) in this version.

Best regards

Reviewer 2 Report

Thank you for the opportunity to discuss your manuscript. All my previous comments were properly addressed. I have no new comments.